# CHUNKED AUTOREGRESSIVE GAN FOR CONDITIONAL WAVEFORM SYNTHESIS

**Max Morrison**[*]
Northwestern University
morrimax@u.northwestern.edu

**Rithesh Kumar, Kundan Kumar**[1] **& Prem Seetharaman**
Descript, Inc.
{rithesh, kundan, prem}@descript.com

**Aaron Courville**[1,2] **& Yoshua Bengio**[1,3]
[1] Mila, Québec Artificial Intelligence Institute, Université de Montréal
[2] CIFAR Fellow
[3] CIFAR Program Co-director

## ABSTRACT

Conditional waveform synthesis models learn a distribution of audio waveforms given conditioning such as text, mel-spectrograms, or MIDI. These systems employ deep generative models that model the waveform via either sequential (*autoregressive*) or parallel (*non-autoregressive*) sampling. Generative adversarial networks (GANs) have become a common choice for non-autoregressive waveform synthesis. However, state-of-the-art GAN-based models produce artifacts when performing mel-spectrogram inversion. In this paper, we demonstrate that these artifacts correspond with an inability for the generator to learn accurate pitch and periodicity. We show that simple pitch and periodicity conditioning is insufficient for reducing this error relative to using autoregression. We discuss the inductive bias that autoregression provides for learning the relationship between instantaneous frequency and phase, and show that this inductive bias holds even when autoregressively sampling large chunks of the waveform during each forward pass. Relative to prior state-of-the-art GAN-based models, our proposed model, Chunked Autoregressive GAN (CARGAN) reduces pitch error by 40-60%, reduces training time by 58%, maintains a fast generation speed suitable for real-time or interactive applications, and maintains or improves subjective quality.

## 1 INTRODUCTION

Conditional audio waveform generation has seen remarkable improvements in fidelity and computational cost in recent years, benefiting applications such as text-to-speech (Kim et al., 2021), voice editing (Qian et al., 2020; Morrison et al., 2020), and controllable music generation (Dhariwal et al., 2020). These advancements are driven by concurrent improvements in deep generative models (Bond-Taylor et al., 2021), which model the distribution of training data with or without auxiliary conditioning information. Conditional audio generation models distributions of audio waveforms with conditioning such as text (Kim et al., 2021), mel-spectrograms (Kong et al., 2020), or MIDI (Hawthorne et al., 2018). In this paper, we focus on tasks such as spectrogram-to-waveform inversion, where the temporal alignment between the conditioning and the waveform is known and the alignment does not have to be learned by the model (Donahue et al., 2020; Kim et al., 2021).

Generative neural networks are the dominant method for audio generation tasks, strongly outperforming comparable methods that use only digital signal processing (DSP) techniques (Griffin & Lim, 1984; Morise et al., 2016). Neural methods learn the waveform distribution directly from

---

[*]This work was carried out during an internship at Descript Inc.

data. Let $\boldsymbol{x} = \{x_1, \ldots, x_T\}$ be an audio waveform and $\boldsymbol{c} = \{c_1, \ldots, c_U\}$ be an aligned—but not necessarily same length—conditioning signal (e.g., a mel-spectrogram). Conditional neural audio generation aims to learn the conditional distribution $p(\boldsymbol{x}|\boldsymbol{c})$. Such models are commonly classified as either generating samples sequentially (*autoregressive*) or in parallel (*non-autoregressive*). We next describe autoregressive and non-autoregressive methods for audio generation, as well as hybrid methods that aim to combine the benefits of both methods.

**Autoregressive methods** Autoregressive models parameterize $p(\boldsymbol{x}|\boldsymbol{c})$ via the following factorization.

$$p(\boldsymbol{x}|\boldsymbol{c}) = \prod_{t=0}^{T} p(x_t|x_1, \ldots, x_{t-1}, \boldsymbol{c}) \tag{1}$$

These models produce subjectively high-quality audio. However, autoregressive generation involves sequentially sampling each audio sample, where each sample requires a forward pass through a neural network. Thus, one major disadvantage of autoregressive models is that generation is slow.

Examples of autoregressive models include WaveNet (Oord et al., 2016), SampleRNN (Mehri et al., 2016), WaveRNN (Kalchbrenner et al., 2018), and Jukebox (Dhariwal et al., 2020). WaveNet introduced the idea of autoregressive waveform generation, demonstrating high-quality generation of speech from aligned linguistic features (e.g., pitch and phonemes) as well as unconditioned generation of speech and music. SampleRNN uses a hierarchy of gated recurrent units (Cho et al., 2014) that operate at increasing temporal resolution to reduce the computation needed to generate each sample. WaveRNN uses a smaller architecture and methods such as weight sparsification (Narang et al., 2017; Zhu & Gupta, 2017) to improve generation speed without sacrificing quality. The resulting model can perform real-time generation, but requires a complex engineering pipeline that includes sparse matrix operations and custom GPU kernels. Jukebox uses a sparse autoregressive transformer (Child et al., 2019) to generate discrete codes of a hierarchical VQVAE (Razavi et al., 2019). Jukebox can generate coherent musical excerpts from a variety of genres with rich conditioning, but requires over eight hours on a V100 GPU to produce one minute of audio.

**Non-autoregressive methods** Non-autoregressive models parameterize $p(\boldsymbol{x}|\boldsymbol{c})$ directly, using methods such as normalizing flows (Dinh et al., 2014), denoising diffusion probabilistic models (DDPMs) (Ho et al., 2020), source-filter models (Fant, 1970), and generative adversarial networks (GANs) (Goodfellow et al., 2014). Exemplar models for conditional audio generation include the flow-based WaveGlow (Prenger et al., 2019), the DDPM-based WaveGrad (Chen et al., 2020), the Neural Source-Filter (NSF) model (Wang et al., 2019), and the GAN-based HiFi-GAN (Kong et al., 2020). WaveGlow is composed of invertible operations with simple Jacobians such as 1x1 convolutions and affine transformations (Kingma & Dhariwal, 2018) to model a sequence of functions of random variables drawn from increasingly complex distributions. While generation with WaveGlow is fast, it requires substantial training resources (e.g., a week of training on 8 V100s) as well as memory resources due to the high parameter count. WaveGrad models the gradient of the log probability density (i.e., the *score function*) via score matching (Hyvärinen & Dayan, 2005) and draws samples from the score-based model using Langevin dynamics (Song & Ermon, 2019). Samples provided by the authors of WaveGrad indicate that the model does not accurately reproduce high-frequency content. NSF creates a deterministic waveform with the correct frequency and uses a neural network to transform this waveform into speech. Source-filter models require accurate pitch estimation and are restricted to monophonic speech signals. HiFi-GAN uses a convolutional generator to generate audio and multiple discriminators that evaluate the generated waveform at various resolutions and strides. HiFi-GAN permits fast generation on CPU and GPU, but requires two weeks of training on two V100 GPUs and regularly produces artifacts, which we discuss in Section 2.

**Hybrid methods** Hybrid methods combine the strengths of both autoregressive and non-autoregressive models by generating more than one sample on each forward pass. Existing hybrid methods for waveform synthesis parameterize $p(\boldsymbol{x}|\boldsymbol{c})$ by factoring into chunks of adjacent audio samples that are autoregressively generated.

$$p(\boldsymbol{x}|\boldsymbol{c}) = \prod_{t=\{k,2k,\ldots,T\}} p(x_{t-k}, \ldots, x_t|x_1, \ldots, x_{t-k-1}, \boldsymbol{c}) \tag{2}$$

Subscale WaveRNN (Kalchbrenner et al., 2018) is one such model, generating 16 samples at a time. WaveFlow (Ping et al., 2020) and Wave-Tacotron (Weiss et al., 2021) are both flow-based

hybrid models. WaveFlow uses autoregression to handle short-range dependencies and a non-autoregressive architecture to model long-range correlations, while Wave-Tacotron uses autoregression for long-range dependencies and a non-autoregressive model for short-range dependencies. These hybrid models demonstrate a tradeoff between the high model capacity of autoregressive models and the low latency of non-autoregressive models.

In this work, we present Chunked Autoregressive GAN (CARGAN), a hybrid GAN-based model for conditional waveform synthesis. CARGAN features fast training, reduced pitch error, and equivalent or improved subjective quality relative to previous GAN-based models. Specifically, we make the following contributions.

- We show that GAN-based models such as HiFi-GAN do not accurately preserve the pitch and periodicity of the audio signal, causing audible artifacts in the generated audio that persist across many recent works.
- We demonstrate a close relationship between pitch, phase, and autoregression, and show that autoregressive models possess an inductive bias towards learning pitch and phase that is related to their ability to learn the cumulative sum operator.
- We show that performing autoregression in large chunks allows us to maintain the inductive bias of autoregression towards modeling pitch and phase while maintaining a fast generation speed, and that the optimal chunk size is related to the causal receptive field of the generator.
- We identify and document common artifacts in GAN-based waveform synthesis, providing examples and offering possible causes. We also make our code available under an open-source license[1]

## 2 PROBLEMS WITH NON-AUTOREGRESSIVE GANS

State-of-the-art GAN-based waveform synthesis models, such as HiFi-GAN (Kong et al., 2020), demonstrate impressive subjective quality and generation speed, but exhibit pitch inaccuracy, audible artifacts caused by periodicity inaccuracy, and limitations induced by the objective function.

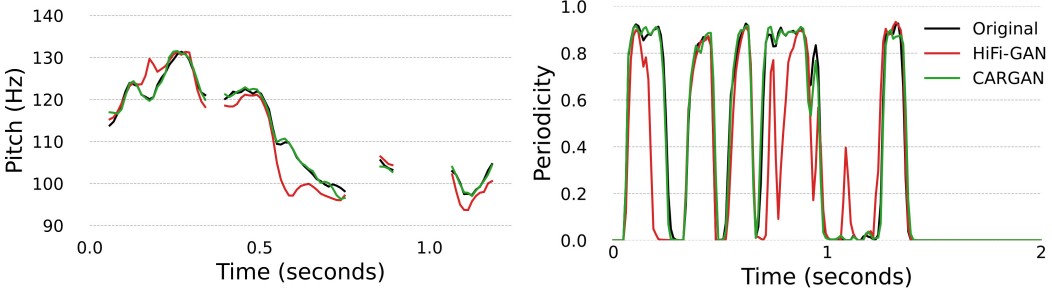

Figure 1: **Left** An example of pitch inaccuracy where HiFi-GAN has an average error of 65.8 cents and CARGAN obtains an error of 15.1 cents. **Right** An example of periodicity inaccuracy where HiFi-GAN has an RMSE of 0.19 and CARGAN obtains an RMSE of 0.04.

**Pitch error artifacts** We trained HiFi-GAN (V1) on the VCTK dataset (Yamagishi et al., 2019) and evaluated the pitch accuracy on 256 randomly selected sentences from a validation set containing speakers seen during training. We use the pitch representation described in Appendix A to extract pitch as well as a binary voiced/unvoiced classification for each frame, which indicates whether the frame exhibits the periodic structure of a pitched sound. We measure the pitch error in regions where both the original and generated speech are voiced. We measure the root-mean-square of the pitch error in cents, defined as $1200 \log_2(y/\hat{y})$ for pitch values $y, \hat{y}$ in Hz. We find an average error of 51.2 cents—more than half a semi-tone. A side-by-side listening comparison of examples with high pitch error and the ground truth audio confirms that current GAN models make audible intonation errors (see Figure 1). Listening examples available on the companion website[2].

---

[1]Code is available at `https://github.com/descriptinc/cargan`.
[2]Audio examples are available at `https://maxrmorrison.com/sites/cargan`.

**Periodicity artifacts** High pitch error indicates inaccurate reconstruction, but is often not perceptible in isolation as speech with slightly different pitch may still sound natural. We manually collect examples from HiFi-GAN that contain audible artifacts. Using our periodicity representation described in Appendix A, which uses the confidence of a pretrained neural pitch estimator to measure whether a frame of audio exhibits periodic structure between zero (completely aperiodic) and one (completely periodic), we find that these artifacts are commonly regions of high periodicity error (see Figure 1). This indicates that existing GAN-based models make audible voiced/unvoiced errors that perceptually degrade the audio. On the companion website, we provide listening examples as well as examples of this artifact from multiple recent works to demonstrate its prevalence. **We strongly encourage readers to listen to these examples, as addressing this artifact is central to our work.**

**Inherent limitations** Current GAN-based models for waveform synthesis depend on a feature matching loss (Larsen et al., 2016; Kumar et al., 2019). This loss is computed by passing the generated and corresponding real audio through the discriminators, and taking the L1 distances between all activations. Penalizing the L1 distance between activations of early layers requires the generator to produce a waveform that is close to ground truth after only one convolution and non-linearity. This requires the generator to know the initial phase. Otherwise, the generator can incur a large loss due to phase rotation. As we will discuss in Section 3, the initial phase is not known in the non-autoregressive setting.

## 3 RELATING AUTOREGRESSION, PITCH, AND PHASE

Our work uses autoregression to address the issues presented in Section 2. We use autoregression to improve the pitch accuracy of HiFi-GAN after many less successful experiments using more intuitive methods such as conditioning the generator on pitch, conditioning the discriminator on pitch, and more, as described in Appendix B.

Autoregression is a sensible choice for addressing the issues presented in Section 2 because of the relationship between pitch and phase in a deterministic, periodic signal. Consider a perfectly periodic signal with instantaneous frequency $\boldsymbol{f} = \{f_1, \ldots, f_T\}$ in Hz and sampling rate $r$. The unwrapped instantaneous phase $\boldsymbol{\phi} = \{\phi_0, \phi_1, \ldots \phi_T\}$ can be autoregressively computed as follows.

$$\phi_t = \phi_{t-1} + \frac{2\pi}{r} f_t \tag{3}$$

The relationship between $\boldsymbol{f}$ and $\boldsymbol{\phi}$ is therefore a cumulative sum operation. Autoregression provides an inductive bias for learning an arbitrary-length cumulative sum operation, which relates the instantaneous frequency and instantaneous phase in deterministic signals. Specifically, incorporating autoregression allows the network to learn $\phi_{t-1}$ and $f_t$ from previous waveform samples, which informs the generator of the ground truth phase during training. Prior autoregressive vocoders have shown that the entropy of the distribution over audio samples learned by the model within voiced regions is low—especially relative to unvoiced regions (Jin et al., 2018). Thus, our assumption of a deterministic, periodic signal is a reasonable approximation in voiced regions.

Equation 3 assumes that a single sample is generated during each forward pass. However, if the generator $G$ is capable of learning a fixed-length cumulative sum of length $k$, we can reduce the number of forward passes by a factor of $k$. As well, the instantaneous frequency can be estimated from the last $n$ samples as long as $n$ is sufficiently large to represent one period of the frequency. This allows us to replace explicit pitch conditioning with conditioning on $n$ previous samples.

$$\phi_t, \ldots, \phi_{t+k} = G(\phi_{t-n-1}, \ldots, \phi_{t-1}) \tag{4}$$

Given a generator $G$, how can we determine the maximum length of a cumulative sum that the generator can learn? First, consider a fully-connected layer with $\ell$ input and output channels. This fully-connected layer can learn a cumulative sum of length $\ell$ when the weights form a upper-triangular matrix of all ones. Next, consider a non-causal convolutional layer with kernel size $m$. Equation 4 is strictly causal, so we use only the causal receptive field of the convolutional layer. This indicates that the maximum length of a learnable cumulative sum is $\lfloor (m+1)/2 \rfloor$. By induction, a convolutional generator $G$ is capable of learning a cumulative sum with length equal to its causal receptive field.

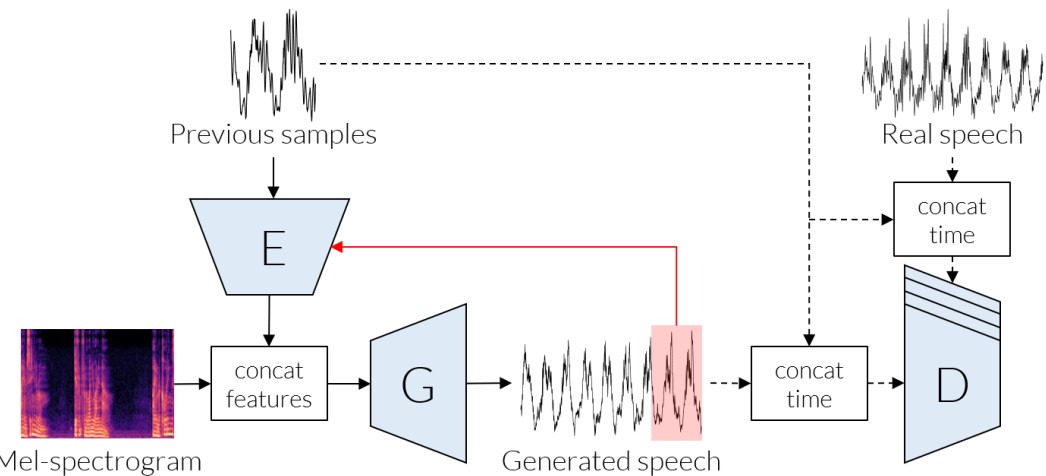

Figure 2: Overview of our proposed Chunked Autoregressive GAN (CARGAN). Blue trapezoids are the learned encoder (E), generator (G), and discriminators (D). Dashed lines represent operations only performed during training. The red line and red highlighted region are the autoregressive loop, where previously generated samples are passed as autoregressive conditioning when generating the next chunk. The previous samples are always prepended when performing concatenation along the time axis.

## 4 CHUNKED AUTOREGRESSIVE GAN

We describe Chunked Autoregressive GAN (CARGAN), our proposed model for conditional waveform synthesis that uses autoregression to address the issues presented in Section 2. CARGAN (Figure 2) consists of three components: (1) an autoregressive conditioning stack, which summarizes the previous $k$ samples into a fixed-length vector, (2) a generator network that converts input conditioning (e.g., a mel-spectrogram) and autoregressive conditioning into a waveform, and (3) a series of discriminators that provide adversarial feedback. The generator and autoregressive conditioning stack are jointly trained to minimize a weighted sum of a differentiable mel-spectrogram loss, the discriminator losses, and the feature matching loss described in Section 2. The discriminators are trained to discern between real and generated audio in a binary classification task.

The previous $k$ samples used for autoregressive conditioning are prepended to the real and generated audio evaluated by the discriminators. This allows the discriminators to evaluate the boundary between the autoregressive and generated audio. Without this step, the generator produces boundary artifacts that sound like periodic clicks. Listening examples available on companion website.

For our generator and discriminator architectures, we use the discriminators from HiFi-GAN (Kong et al., 2020) as well as a modified version of the generator from GAN-TTS (Bińkowski et al., 2019), both of which are described in Appendix C. We find that this modified GAN-TTS generator provides superior performance over the generator of HiFi-GAN regardless of whether autoregression is used.

### 4.1 AUTOREGRESSIVE CONDITIONING STACK

The autoregressive conditioning stack converts the previous $k$ samples of the waveform into a fixed-length embedding. We use a simple architecture, consisting of five linear layers. We use leaky ReLU with a slope of $-0.1$ between layers. The hidden size of all layers is 256 and the output size is 128. The output embedding is concatenated to each frame of the spectrogram that is passed as input to the generator (i.e., the embedding is repeated over the time dimension).

We also experimented with multiple fully convolutional architectures, including convolutional architectures with more layers, a larger capacity, and residual connections (He et al., 2016). We found that the simple fully-connected model outperforms convolutional models. We hypothesize that fully-connected layers are advantageous because they provide an inductive bias towards learning the autocorrelation function, which is a common operation used in DSP-based pitch estima-

tion (De Cheveigné & Kawahara, 2002; Rabiner & Schafer, 1978). While a stack of convolutional layers can learn an autocorrelation, the autocorrelation function cannot be modeled by a single convolution with kernel size smaller than the signal length. However, autocorrelation can be learned with a single fully-connected layer with equal-sized input and output.

## 4.2 Loss compensation

The generator is trained using both feature matching and mel-spectrogram losses. It is important that the ratio of these loss terms remains balanced. Otherwise, artifacts occur. When training with autoregressive conditioning, the feature matching loss decreases significantly relative to the non-autoregressive setting. We find that autoregression also improves other phase-dependent metrics not used as training losses, such as the L2 loss between the generated and ground-truth waveform and the squared error between the phase components of the signal, where each bin of the phase spectrum is weighted by the corresponding magnitude. This indicates that the autoregressive conditioning stack is successfully providing phase information to the generator. However, with the feature matching loss reduced, the mel-spectrogram error dominates the training criteria. This produces a metallic sound in the audio, especially during breath sounds and unvoiced fricatives of speech data. For example, /ʃ/ (e.g., "she") ends up sounding closer to the corresponding voiced fricative /ʒ/ (e.g., "Asia"). Listening examples available on the companion website. We fix these metallic artifacts by rebalancing the losses: while HiFi-GAN originally uses weights of 2 and 45 for the feature matching and mel-spectrogram losses, respectively, we use 7 and 15.

## 5 Experiments

The following sections describe experiments that evaluate our proposed claims. First, we will show that autoregressive models are more capable than non-autoregressive models at learning an arbitrary length cumulative sum (Section 5.1). Next, we perform spectrogram-to-waveform inversion on speech (Section 5.2), and show that CARGAN produces better pitch accuracy, better subjective quality, faster training time, and reduced memory consumption during training. In Appendix J, we also perform spectrogram-to-waveform inversion on music data. We show that CARGAN is capable of modeling reverberant signals, exhibits greater pitch accuracy on musical data, and is more capable of modeling accurate low-frequency information relative to HiFi-GAN.

All models are trained with a batch size of 64. We use the AdamW optimizer (Loshchilov & Hutter, 2017) with a learning rate of $2 \times 10^{-4}$ and $\beta = (.8, .99)$. We use an exponential learning rate schedule that multiplies the learning rate by .999 after each epoch. **For all tables, ↑ means higher is better and ↓ means lower is better.**

## 5.1 Synthetic cumulative sum experiment

Here we show that CARGAN is more capable of modeling an arbitrary-length cumulative sum than HiFi-GAN. We generate a synthetic dataset by replacing each audio sample in the VCTK dataset (Yamagishi et al., 2019) with a random sample taken from a uniform distribution between zero and one. We take the cumulative sums of these random vectors, and normalize both the random vector and cumulative sum by the sum of the random vector. The random vectors are the inputs to the model, and their cumulative sums are the ground truth training targets.

To train on this synthetic dataset, we must modify the generator architecture and training criteria. Given that the input and target sequences have the same sampling rate, we remove all upsampling operations from the generator. We reduce the number of channels in each layer by a factor of four to reduce training time. The target sequences share a deterministic relationship with the input. As such, we replace the mel error, adversarial, and feature matching losses with a simple L1 loss.

The autoregressive conditioning of CARGAN informs the model of the current total of the cumulative sum. Because HiFi-GAN does not have access to this information, we train HiFi-GAN using only training data where the current total is zero. This can be viewed as subtracting the current total from the training target prior to non-autoregressive training.

We train each model for 100,000 steps and evaluate the L1 distance on 256 held-out examples. We train CARGAN with a chunk size of 2048 and use 512 previous samples for autoregressive

conditioning. We perform evaluation for signal lengths of 1024, 2048, 4096, 8192, and "Full", where "Full" indicates that the entire available signal length is used. We include two additional conditions. The first is HiFi-GAN with our modified GAN-TTS generator (**+ GAN-TTS**) to demonstrate that our improvements on this synthetic task are primarily due to autoregression, and not the larger causal receptive field of the generator. The second is CARGAN with a larger kernel size of 15 in all convolution layers (**+ Large kernel**) to show the impact of a larger causal receptive field in the autoregressive case.

Results are presented in Table 1. We find that CARGAN provides substantial improvement in the L1 error for all lengths relative to HiFi-GAN. The non-autoregressive models were trained with a sequence length of 8192 and overfit to that sequence length, leading to reduced accuracy even for small sequence lengths. Specifically, Figure 3 shows that the non-autoregressive model converges to the mean of the training data, whereas the CARGAN converges to the piece-wise mean for each chunk. When a larger kernel is used, CARGAN has a causal receptive field greater than the chunk size. This allows CARGAN to learn a smooth interpolation of each chunk and model very long cumulative sums—consistent with our analysis in Section 3. Note that this analysis does not include upsampling operations in the generator, which also impacts the size of the causal receptive field in tasks such as spectrogram inversion.

| Method | Causal Receptive Field | 1024 | 2048 | 4096 | 8192 | Full |
|---|---|---|---|---|---|---|
| HiFi-GAN | 245 | .050 | .042 | .028 | .031 | .447 |
| + GAN-TTS | 402 | .052 | .042 | .028 | .029 | .447 |
| CARGAN | 402 | **.009** | .015 | .021 | .025 | .359 |
| + Large kernel | 2802 | **.009** | **.013** | **.019** | **.024** | **.132** |

Table 1: Results of our synthetic cumulative sum experiment. All values are L1 distances between the predicted and ground truth cumulative sums.

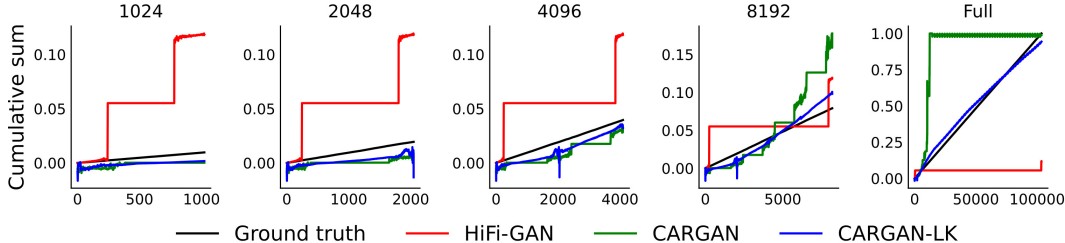

Figure 3: Example output of our synthetic cumulative sum experiment at various lengths. CARGAN-LK is our proposed model with a larger kernel size.

## 5.2 SPECTROGRAM-TO-WAVEFORM

We perform spectrogram-to-waveform inversion on speech. We compute the mel-spectrogram from the ground truth audio, and train models to recover the audio waveform. The exact steps for computing our mel-spectrogram representation are detailed in Appendix D. All audio is sampled at a sampling rate of 22050 Hz. Audio with a maximum absolute value of less than .35 is peak-normalized to .35. All models are trained for 500,000 steps. HiFi-GAN was originally trained for 2.5 million steps. However, this takes about one month on one RTX A6000, prohibiting fast experimentation. In Appendix E we justify using 500,000 steps by training CARGAN and HiFi-GAN to 2.5 million steps and showing a comparable gap in pitch and periodicity errors.

We train on the VCTK dataset (Yamagishi et al., 2019) and evaluate on both VCTK and DAPS (Mysore, 2014). For training on VCTK, we randomly select 100 speakers. We train on a random 95% of the data from these 100 speakers, using data from both microphones. For evaluation on DAPS, we use the segmented dataset of the first script of the clean partition available on Zenodo (Morrison et al., 2021).

We use a chunk size of 2048 samples and an autoregressive conditioning input size of 512 samples in our proposed CARGAN model. In other words, the model generates 2048 samples on each forward

| Method | VCTK | | | DAPS | | |
|---|---|---|---|---|---|---|
| | Pitch↓ | Periodicity↓ | F1↑ | Pitch↓ | Periodicity↓ | F1↑ |
| HiFi-GAN | 51.2 | .113 | .941 | 54.7 | .142 | .942 |
| CARGAN | 29.4 | **.086** | **.956** | **21.6** | **.107** | **.959** |
| - GAN-TTS | 37.9 | .099 | .949 | 27.0 | .117 | .953 |
| - Loss balance | 33.7 | .104 | .943 | 34.1 | .119 | .952 |
| - Prepend | **24.6** | .088 | .955 | 24.4 | .108 | .958 |

Table 2: Objective evaluation results for spectrogram-to-waveform inversion on VCTK and DAPS.

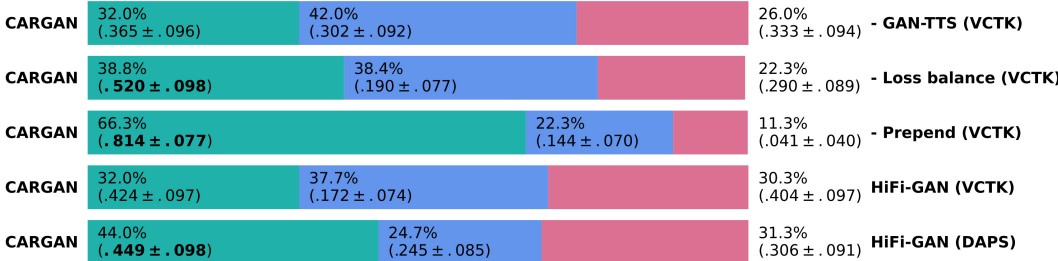

Figure 4: Subjective pairwise test results for audio quality and naturalness on VCTK and DAPS. Each row contains the percent preference between two methods (including ties) as well as Bernoulli confidence intervals (CIs) for the probability of our method being better, equal to, or worse than the competing method at a *p*-value of 0.05. Bolded CIs indicate a statistically significant winning method. Ablations are defined in Section 5.2.

pass, and passes the last 512 generated samples as autoregressive conditioning for the next forward pass. In contrast, HiFi-GAN produces 8192 samples on each forward pass during training, but many more samples during generation. The optimal chunk size is selected based on the hyperparameter search described in Appendix F. As the number of previous samples passed as autoregressive conditioning increases, we notice two behaviors: (1) it lowers the minimal representable frequency that can be modeled and (2) it significantly increases training time, as the discriminators—which dominate training time—must also process the full autoregressive conditioning. We approximate the minimum pitch of typical human speech as 50 Hz, which corresponds to a wavelength of 441 samples given a sampling rate of 22,050 Hz. Thus, 512 is the smallest power of two capable of representing at least one cycle of a 50 Hz waveform.

We perform both objective and subjective evaluation. For the objective evaluation, we measure pitch error (in cents), periodicity RMSE, and voiced/unvoiced classification F1 score. We use the pitch and periodicity representations described in Appendix A. Objective evaluation is performed on 256 randomly selected held-out examples from VCTK and DAPS. For VCTK, we select examples from seen speakers not used during training. For subjective evaluation, we use Amazon Mechanical Turk to perform pairwise tests between our proposed CARGAN and the baseline HiFi-GAN on both VCTK and DAPS. In each pairwise test, 15 participants listen to 20 randomly selected pairs and select which example sounds best based on speech naturalness and audio quality. Participants are also given the option of indicating that both examples exhibit equal quality and naturalness. We construct pairs from 100 randomly selected examples from each dataset. For VCTK, we additionally perform three pairwise ablation tests to measure the importance of our proposed methods: (1) **- GAN-TTS** we use the generator from HiFi-GAN instead of GAN-TTS, (2) **- Loss balance** we omit the loss compensation of Section 4.2, using the original weights of 2 and 45 for the mel-spectrogram and feature matching losses, respectively, and (3) **- Prepend** we omit prepending the autoregressive conditioning to the signals passed into the discriminator, so that the discriminator is unable to see the boundary between the autoregressive conditioning and the generated or real continuation.

Results for objective evaluation of our spectrogram-to-waveform experiments on VCTK and DAPS data are presented in Table 2. Figure 4 shows the results of the subjective listening tests on both datasets. For the objective evaluation, we find that CARGAN substantially reduces the pitch and periodicity error and increases the F1 score of voiced/unvoiced classification. This corroborates our

hypothesis presented in Section 3 that autoregression provides an inductive bias for learning accurate pitch and phase information. While our **- Prepend** ablation exhibits slightly better pitch accuracy on VCTK, subjective evaluation demonstrates that the boundary artifacts overwhelmingly degrade the audio signal. For subjective evaluation, we find that CARGAN exhibits equal quality to HiFi-GAN on VCTK, and superior quality on DAPS. CARGAN also ties with our **- GAN-TTS** ablation; however, the objective evaluation indicates that CARGAN exhibits superior pitch and periodicity error, as well as F1, relative to this ablation. We also explored increasing the causal receptive field of the generator by increasing the kernel size of all convolutional layers to 15, as in Section 5.1. We obtain a small improvement in pitch error (24.3 and 20.3 cents on VCTK and DAPS, respectively), but no improvement in periodicity error or F1 and a significant reduction in generation speed.

In Appendix G we report the DeepSpeech distances proposed in Bińkowski et al. (2019). We do not find that these experimental metrics correlate well with our objective or subjective metrics. In Appendix H, we show that there is a weak correlation between subjective preference and periodicity RMSE, and a smaller correlation between subjective preference and pitch RMSE. This corroborates our hypothesis in Section 2 that intonation errors are not as perceptible as periodicity errors.

## 5.3 TRAINING AND GENERATION SPEED

| Method | Training | | Generation | |
|---|---|---|---|---|
| | Speed (ms/step)↓ | Memory use (GB)↓ | GPU (RTF)↑ | CPU (RTF)↑ |
| HiFi-GAN | 1186 | 40.5 | **180.8** | **1.21** |
| CARGAN | **502** | **12.4** | 10.7 | 0.45 |

Table 3: Results for time and memory benchmarking of HiFi-GAN and CARGAN. Real-time factor (RTF) is the number of seconds of audio that can be generated per second.

We benchmark HiFi-GAN and CARGAN to determine the relative speed and memory consumption when performing spectrogram-to-waveform inversion on speech data (Section 5.2). We use a single RTX A6000 for training and generation on a GPU, and two cores of an AMD EPYC 7742 with one thread per core and a 2.25 GHz maximum clock speed for CPU benchmarking.

Results of our benchmarking are presented in Table 3. We find that CARGAN reduces training time by 58% and reduces memory consumption during training by 69%. These improvements are due to the large reduction in training sequence length from 8192 in HiFi-GAN to 2048 in CARGAN. While generation is slower with CARGAN, we can easily improve generation speed at the cost of reduced training speed, increased memory usage, and slightly increased pitch error by changing the chunk size (see Appendix F). As well, the autoregressive nature of CARGAN makes it suitable for streaming-based applications running over a low-bandwidth network, as not all features have to be available to begin generation (i.e., the conditioning in Equation 2 is also factorized to be causal).

## 6 CONCLUSION

In this paper, we proposed Chunked Autoregressive GAN (CARGAN), a GAN-based model for conditional waveform synthesis. Relative to existing methods, CARGAN demonstrates improved subjective quality and pitch accuracy, while significantly reducing training time and memory consumption. We show that autoregressive models permit learning an arbitrary length cumulative sum operation, which relates the instantaneous frequency and phase of a periodic waveform. We demonstrate that the benefits of autoregressive modeling can be realized while maintaining a fast generation speed by generating large chunks of audio during each forward pass, where the optimal chunk size is related to the causal receptive field of the generator architecture.

Our work reveals multiple directions for future improvement. The generation speed of CARGAN is largely determined by the chunk size. Designing generators with a large causal receptive field may further improve generation speed by permitting generation in larger chunks. As well, while CARGAN addresses artifacts caused by pitch and periodicity errors, it induces occasional boundary artifacts in the signal—even when the discriminator is able to evaluate the boundary between autoregressive conditioning and the generated or real continuation. Future work that addresses this artifact will further improve subjective quality.

**Acknowledgments** The authors would like to thank Jose Sotelo, Lucas Gestin, Vicki Anand, and Christian Schilter for valuable discussions and inputs.

**Ethics statement** Our work involves a human subject study, utilizes standard speech datasets, and produces a system that can be incorporated into a voice cloning system. Our human subject study is conducted with care. We use standard speech datasets that are free of profane or objectionable content. We also include instructions that encourage participants to set their volume levels at a reasonable level to prevent hearing loss. The datasets we use are common in the speech research community, but are biased toward American and British English. We note the need for clean speech datasets that contain proportional or equal representations of accents and dialects across the globe. Finally, our system could be used as one component of a voice cloning system. While such systems have profound utility in podcast and film dialogue editing, they can also be used to mimic one's voice without consent. We do not condone the use of our work for non-consensual voice cloning.

**Reproducibility** In order to facilitate reproduction of our research, we provide documented, open-source code that permits reproducing and evaluating all experiments in our paper, except for pairwise subjective evaluation on Amazon Mechanical Turk, which uses proprietary code. We also provide a pretrained model, a script and library for performing generation, and packaging via PyPi and PyTorch Hub.

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

## A   PITCH REPRESENTATION

We use `torchcrepe` (Morrison, 2020) to extract pitch and periodicity features from an audio waveform. `torchcrepe` first predicts a categorical distribution over possible pitch values using CREPE (Kim et al., 2018), a neural pitch estimator trained on music data. We set the probability of any pitch values outside the human speaking range (approximately 50-550 Hz) to zero and normalize the resulting distribution to sum to one. This produces a sequence of categorical distributions over each frame of audio. We perform Viterbi decoding on this sequence of distributions to determine the maximum likelihood path. For Viterbi transition probabilities, we use a triangular distribution that assigns zero probability to pitch jumps greater than one octave between adjacent frames. Each step along the maximum likelihood path has an index associated with a pitch value as well as a probability value predicted by the model. We convert the path indices to frequencies in Hz to produce a pitch contour, and use the sequence of probability values as the periodicity contour. Because CREPE is invariant to amplitude, low-bit noise during nearly silent regions can induce high periodicity. We compute the A-weighted loudness (McCurdy, 1936) at each frame of audio, and set the periodicity to zero in frames where the loudness is less than -60 dB relative to a reference level of 20 dB.

CREPE independently predicts a pitch distribution for 1024 samples of audio at a sampling rate of 16000 Hz. Our audio data is sampled at 22050 kHz, and our mel spectrogram features are computed with a hop size of 256 samples. To align pitch features with the mel spectrogram, we replace the default padding of CREPE with the same reflection padding used for mel spectrogram computation (Section 5.2) and run CREPE with a hop size of $256 \times 16000/22050 = 185.8$ samples. `torchcrepe` does not currently support a fractional hop size. We instead use a hop size of 185 samples and linearly resample to the target length if needed. Resampling of pitch values is performed in base-2 log-space.

Given the pitch and periodicity contour of a speech signal, we can extract a binary value for each frame that indicates whether or not the frame contains a voiced phoneme. Intuitively, when the entropy of the distribution produced by CREPE is low, the signal is likely to be periodic with a frequency given by the peak of the distribution (i.e., the periodicity will be high). While one could perform this voiced/unvoiced classification by simply thresholding the periodicity at a given value, we find that this induces *chatter*, which occurs when a signal oscillates above and below a threshold. We propose to use *hysteresis* thresholding, which requires the signal to remain above the threshold for a certain number of frames in order to be considered voiced. This hysteresis thresholding is also how chatter is reduced in audio compressors (Senior, 2011).

## B   PITCH CONDITIONING EXPERIMENTS

Our work proposes to use autoregression to improve the pitch and periodicity error of GAN-based speech synthesis. However, autoregression is not an intuitive solution for improving pitch and periodicity error compared to, e.g., conditioning on pitch features. During development, we also explored a number of these more intuitive methods. Here we describe a subset of the experiments we tried. All experiments are performed using the baseline HiFi-GAN model.

**Generator conditioning** We condition the generator on the pitch and periodicity features described in Appendix A. These are concatenated to the input mel-spectrogram features as two additional channels. Pitch features are converted to base-2 log-space prior to concatenation.

**Discriminator conditioning** We condition the discriminator on pitch and periodicity features. We linearly upsample both the pitch (in base-2 log-space) and periodicity features to the same temporal resolution as the generated waveform, and concatenate the upsampled pitch and periodicity as two additional channels. Intuitively, this allows the discriminator to determine if the pitch and periodicity of the generated speech is consistent with the ground truth pitch and periodicity features.

**CREPE perceptual loss** We pass the real and generated audio into the CREPE pitch estimator (Kim et al., 2018) and perform feature matching between all activations, including the unnormalized logits. The weights of CREPE are frozen during training. We compute the L1 loss between all activations and add the result to the generator loss of HiFi-GAN.

**Pitch discriminator** Inspired by the OASIS model for semantic image segmentation (Schönfeld et al., 2020), we design a discriminator that jointly performs pitch estimation and real/fake classification. We use the architecture of CREPE for the discriminator. Our pitch-estimating discriminator predicts a 256-dimensional categorical distribution, where the first 255 categories represent evenly quantized pitch values between 50 and 550 Hz in base-2 log-space, and the final category is used by the discriminator to indicate that the input audio is fake. The discriminator is trained to categorize all real data as its ground truth pitch and all generated data as fake. The generator is trained to produce audio that is categorized as the ground truth pitch by the pitch-estimating discriminator. These generator and discriminator losses are added to the generator and discriminator losses of the baseline HiFi-GAN.

| Method | VCTK | | | DAPS | | |
|---|---|---|---|---|---|---|
| | Pitch↓ | Periodicity↓ | F1↑ | Pitch↓ | Periodicity↓ | F1↑ |
| HiFi-GAN | 51.2 | .113 | .941 | 54.7 | .142 | .942 |
| Generator conditioning | 39.1 | .108 | .943 | 49.7 | .135 | .945 |
| Discriminator conditioning | 33.9 | .096 | .951 | 43.0 | .121 | .952 |
| CREPE perceptual loss | 53.6 | ,101 | .948 | 54.6 | .137 | .948 |
| Pitch discriminator | 48.9 | .109 | .943 | 53.4 | .139 | .943 |
| CARGAN | **29.4** | **.086** | **.956** | **21.6** | **.107** | **.959** |

Table 4: Results for objective evaluation of non-autoregressive pitch conditioning relative to the baseline HiFi-GAN and our proposed, autoregressive CARGAN.

We provide the pitch and periodicity error for these four experiments, as well as the baseline HiFi-GAN, on the same 256 examples from the DAPS dataset as used for evaluation in Section 5.2. Results are presented in Table 4. These results indicate that autoregression plays an important role in reducing pitch and periodicity error that cannot be fully compensated for by more intuitive methods in the non-autoregressive case.

## C  NETWORK ARCHITECTURE

| Layer | Input channels | Output channels | Upsampling ratio |
|---|---|---|---|
| 1 | 768 | 768 | 1 |
| 2 | 768 | 768 | 1 |
| 3 | 768 | 384 | 4 |
| 4 | 384 | 384 | 4 |
| 5 | 384 | 384 | 4 |
| 6 | 384 | 384 | 1 |
| 7 | 384 | 192 | 2 |
| 8 | 192 | 192 | 1 |
| 9 | 192 | 96 | 2 |
| 10 | 96 | 96 | 1 |

Table 5: Layer configuration for all GBlocks in our modified GAN-TTS generator.

Our modified GAN-TTS generator consists of a 1x1 convolution followed by a sequence of GBlocks and an output convolution with tanh activation. All convolutional layers in the generator have a kernel size of three and are padded to maintain the same temporal dimension unless otherwise speci-

fied. Each GBlock contains two convolutional blocks. The first block consists of a ReLU activation, optional nearest neighbors upsampling, a convolutional layer, ReLU activation, and a dilated convolutional layer with a dilation rate of three samples. The second block consists of a ReLU activation, dilated convolution with a dilation rate of nine, a second ReLU, and a second dilated convolution with a dilation rate of 27. The input to the generator is passed through the first convolutional block as well as a residual path that consists only of optional upsampling and a 1x1 convolution. The output of the first block and the residual connection are added and passed through the second convolutional block. A residual connection adds the input and output of the second convolutional block. Table 5 provides the number of channels and ratio of nearest neighbors upsampling for each GBlock in our generator.

We use the same discriminators as HiFi-GAN (Kong et al., 2020). HiFi-GAN utilizes eight discriminators to provide adversarial feedback to the generator. Three of these discriminators are multi-scale discriminators (MSDs), each of which evaluates waveforms at a different resolution: raw audio, 2x average-pooled audio, and 4x average pooled-audio. Each MSD consists of eight grouped and strided 1D convolutional layers with Leaky ReLU activations. The MSD that operates on raw audio uses spectral normalization (Miyato et al., 2018), while the other two use weight normalization (Salimans & Kingma, 2016). The other five discriminators are multi-period discriminators (MPDs), each of which evaluates the input audio of length $T$ by first reshaping into 2D matrices of dimension $T/p \times p$ for a different prime number $p \in [2, 3, 5, 7, 11]$. Each MPD consists of six strided 2D convolutions with weight normalization and Leaky ReLU activations.

## D   MEL-SPECTROGRAM REPRESENTATION

To compute the mel-spectrogram, we first compute the magnitude of the STFT of the waveform. We map each frame of the magnitude spectrogram onto the mel scale using a triangular filterbank. We clamp the resulting mel energies to have a minimum of $1 \times 10^{-5}$ and take the base-10 log. We use 1024 frequency channels for the STFT, with a window size of 1024 samples and a hop size of 256 samples. We use 80 frequency channels for our mel spectrogram representation. Prior to computing the STFT, we pad the audio using reflection padding so that the real and generated waveforms are equal length when the real audio is divisible by the hop size.

## E   PITCH AND PERIODICITY ERROR AT 2.5 MILLION STEPS

| | VCTK | | | DAPS | | |
|---|---|---|---|---|---|---|
| Method | Pitch↓ | Periodicity↓ | F1↑ | Pitch↓ | Periodicity↓ | F1↑ |
| HiFi-GAN (0.5M) | 51.2 | .113 | .941 | 54.7 | .142 | .942 |
| HiFi-GAN (2.5M) | 61.2 | .094 | .950 | 43.5 | .124 | .952 |
| CARGAN (0.5M) | **29.4** | .086 | .956 | **21.6** | .107 | .959 |
| CARGAN (2.5M) | 31.7 | **.077** | **.961** | 22.6 | **.090** | **.966** |

Table 6: Pitch and periodicity error for a varying number of steps

We compute the pitch and periodicity error of HiFi-GAN and CARGAN at 0.5 million and 2.5 million steps. For HiFi-GAN, training to 2.5 million steps takes about one month on an RTX A6000 GPU. We use the same 256 examples from VCTK and DAPS as used for objective evaluation in Section 5.2. Results are presented in Table 6. We see that CARGAN outperforms HiFi-GAN at both 0.5 million and 2.5 million steps. Given the shorter training time and proportional pitch and periodicity errors, we use 0.5 million steps for all other experiments.

## F   CHUNK SIZE ABLATION

Our proposed autoregressive model, CARGAN, generates 2048 samples during each forward pass. Here we justify using 2048 samples on each forward pass by comparing the pitch and periodicity error as well as training and generation speeds at various chunk sizes. We train our best model using chunk sizes of 512, 1024, 2048, 4096, and 8192 and evaluate using the same 256 samples from DAPS as used for evaluation in Section 5.2. Results are presented in Table 7. We find that a chunk

| Chunk size | Objective metrics | | | Speed | |
|---|---|---|---|---|---|
| | Pitch↓ | Periodicity↓ | F1↑ | Training (ms/step)↓ | GPU Generation (RTF)↑ |
| 512 | 22.9 | .109 | .956 | **269** | 2.43 |
| 1024 | 22.1 | .112 | .955 | 354 | 4.77 |
| 2048 | **21.6** | **.107** | **.959** | 502 | 10.7 |
| 4096 | 24.8 | .111 | .957 | 821 | 20.5 |
| 8192 | 24.1 | .108 | **.959** | 1411 | **39.4** |

Table 7: Results for objective evaluation and benchmarking of CARGAN using various chunk sizes. Real-time factor (RTF) is the number of seconds of audio that can be generated per second.

size of 2048 is optimal for pitch, periodicity and F1. As the chunk size increases, training speed decreases and generation speed increases. In informal subjective evaluation, we find that smaller chunk sizes are more likely to introduce boundary artifacts (see Section 4), while larger chunk sizes are more likely to introduce periodicity artifacts discussed in Section 2.

## G  DEEPSPEECH OBJECTIVE METRICS

| Method | FDSD↓ | cFDSD↓ | KDSD↓ | cKDSD↓ |
|---|---|---|---|---|
| HiFi-GAN | 4.03 | **.520** | $5 \times 10^{-5}$ | $\mathbf{-14 \times 10^{-5}}$ |
| CARGAN | 4.05 | .778 | $7 \times 10^{-5}$ | $-11 \times 10^{-5}$ |
| - GAN-TTS | 4.04 | .776 | $7 \times 10^{-5}$ | $-11 \times 10^{-5}$ |
| - Loss balance | **4.01** | .554 | $\mathbf{4 \times 10^{-5}}$ | $-13 \times 10^{-5}$ |
| - Prepend | 4.07 | .875 | $13 \times 10^{-5}$ | $-6 \times 10^{-5}$ |

Table 8: DeepSpeech distances on the VCTK dataset

| Method | FDSD↓ | cFDSD↓ | KDSD↓ | cKDSD↓ |
|---|---|---|---|---|
| HiFi-GAN | **3.65** | **.432** | $\mathbf{2 \times 10^{-5}}$ | $\mathbf{-10 \times 10^{-5}}$ |
| CARGAN | 3.69 | .644 | $7 \times 10^{-5}$ | $-6 \times 10^{-5}$ |

Table 9: DeepSpeech distances on the DAPS dataset

We report the Fréchet DeepSpeech distance (FDSD), the Kernel DeepSpeech distance (KDSD) and their conditional variants (cFDSD and cKDSD) (Bińkowski et al., 2019) on HiFi-GAN, CARGAN, and all ablations described in Section 5.2. These are objective metrics inspired by the Fréchet Inception Distance (Heusel et al., 2017) and are meant to approximate subjective preference by comparing distances between embeddings of real and generated audio produced by the DeepSpeech 2 speech recognition system (Amodei et al., 2016). We use the public implementation of these distances published by the original authors. This implementation requires at least 10,000 samples per condition. To obtain these samples, we sample 40 random one-second chunks from each of the 256 examples from the VCTK and DAPS datasets used for objective evaluation. Results on VCTK are presented in Table 8 and results on DAPS are presented in Table 9. Note that the public implementation reports a limited precision for KDSD and cKDSD. Comparing these results to our subjective evaluation in Table 4, we find that the only statistically significant comparison that is also reflected in the DeepSpeech distances is between CARGAN and the **- Prepend** ablation. Therefore, the DeepSpeech distances do not seem to be an adequate indicator of subjective quality when the conditions are as close in subjective quality and naturalness as the ones considered in this paper.

## H  CORRELATION BETWEEN SUBJECTIVE PREFERENCE AND PITCH AND PERIODICITY

We analyze the subjective pairwise results between HiFi-GAN and CARGAN described in Section 5.2. We consider a loss (i.e., the participant thinks HiFi-GAN sounds better) to have a value of 0, a tie as 0.5, and a win (i.e., the participant thinks CARGAN sounds better) as 1. We compute a

score for each example by averaging over all losses, wins, and ties, and compute the Pearson correlation between this score and the RMSE gap between HiFi-GAN and CARGAN. On VCTK, this provides an insignificant correlation of 0.065 with a two-tailed $p$-value of 0.531. The correlation is more pronounced on DAPS, with a value of 0.257 and a $p$-value of 0.011. The correlations for pitch RMSE are significantly less ($-0.187$ with a $p$-value of 0.068 on VCTK and $-0.005$ with a $p$-value of 0.962 on DAPS), which corroborates our hypothesis in Section 2 that intonation errors are not as perceptible as periodicity errors. These correlations also indicate that the artifacts associated with periodicity errors described in Section 2 are not the only source of variability in the reconstructed signal that affect periodicity RMSE.

## I    APPROXIMATE SPEED COMPARISON WITH OTHER MODELS

We provide an approximate comparison of the inference speed of CARGAN relative to WaveNet, WaveGlow, MelGAN, and HiFi-GAN (see Table 10). To compute speed values for CARGAN, we assume linear relationship in both CPU and GPU computation speed between the benchmarking conditions used in HiFi-GAN (Kong et al., 2020) and the conditions described in Section 5.3, where the slope is computed via the relative speeds of HiFi-GAN between benchmarks. This means that the values for CARGAN are not the result of properly controlled benchmarking, and should only be used as an illustrative guide.

| Method | GPU (RTF)↑ | CPU (RTF)↑ |
|---|---|---|
| HiFi-GAN | 167.900 | 1.43 |
| CARGAN | 9.937 | 0.53 |
| WaveNet (MoL) | 0.003 | – |
| WaveGlow | 22.800 | 0.21 |
| MelGAN | 645.730 | 6.59 |

Table 10: Approximate time benchmarking of some recent waveform synthesizers. Real-time factor (RTF) is the number of seconds of audio that can be generated per second. Values other than CARGAN are from Table 1 in the original HiFi-GAN paper.

## J    MUSIC EXPERIMENTS

To show how our model can be adapted to domains other than speech, we apply our proposed model to spectrogram inversion of music. We train on a random 80% of the stems of the MUSDB-HQ dataset (Rafii et al., 2019). If we use the same hyperparameters as we used for speech data for CAR-GAN (a chunk size of 2048 with 512 previous samples used as autoregressive conditioning) we find that the resulting audio contains significant degradations. We hypothesize that this is due to the music dataset containing low frequencies, long reverberation, and polyphony. To handle lower frequencies and long reverberation, we increase the number of previous samples used as conditioning to 16384 samples, which greatly extends the receptive field. Increasing the size of the autoregressive conditioning decreases the feature matching loss. We increase the weight on the feature matching loss from 7 to 21 to compensate. We call this model CARGAN-16k.

We provide listening examples on held-out data from MUSDB-HQ on the companion website. We also design three simple experiments to evaluate the ability of HiFi-GAN, CARGAN, and CARGAN-16k to model low frequencies, reverb, and polyphony. To probe the ability of the models to learn accurate low-frequency information, we pass as input a repeated kick drum sample while gradually increasing the center frequency of a high-pass filter. We repeat the kick drum sample four times. We turn off the high-pass filter on the first repetition, and thereafter use center frequencies of 30, 60, and 90 Hz. The Q-factor of the filter is fixed at 1. In Figure 5, we see that HiFi-GAN ignores the high-pass filter and overemphasizes low frequencies, while CARGAN and CARGAN-16k exhibit low-frequency energy closer to ground truth.

To test the ability of the models on reverberant audio, we use a repeated snare drum sample with gradually increasing decay time. We repeat the snare sample five times. We turn off the reverb on the first repetition, and thereafter use decay times of 250 ms, 500 ms, 1 second, and 2 seconds. In Figure 6, we see that all models are capable of modeling the reverb in this simple example. How-

ever, HiFi-GAN and CARGAN both overemphasize the high frequencies of the transient relative to CARGAN-16k and the ground truth audio.

To test the ability of the model to generate polyphonic audio, we use a MIDI piano instrument playing a C chord with a gradually increasing number of notes. In Figure 7, we see that all models are capable of generating polyphonic audio. However, they exhibit different artifacts: HiFi-GAN exhibits a wide vibrato that indicates low pitch accuracy, while CARGAN and CARGAN-16k exhibit boundary artifacts that appear as repeated clicks. Audio for all experiments in this section can be found on the companion website.

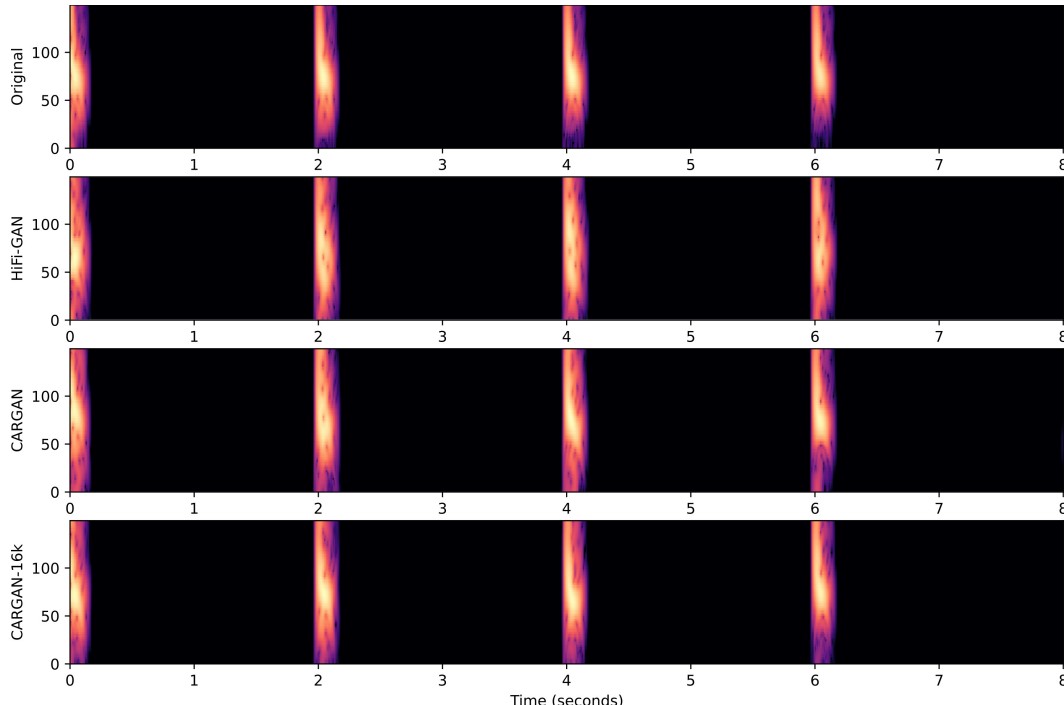

Figure 5: Spectrogram visualizations for our low frequency modeling experiment on a repeated kick drum sample with a gradually increasing center frequency on a high-pass filter. Comparing the lowest frequencies of each transient shows that CARGAN-16k is most capable of modeling the effect of this high-pass filter.

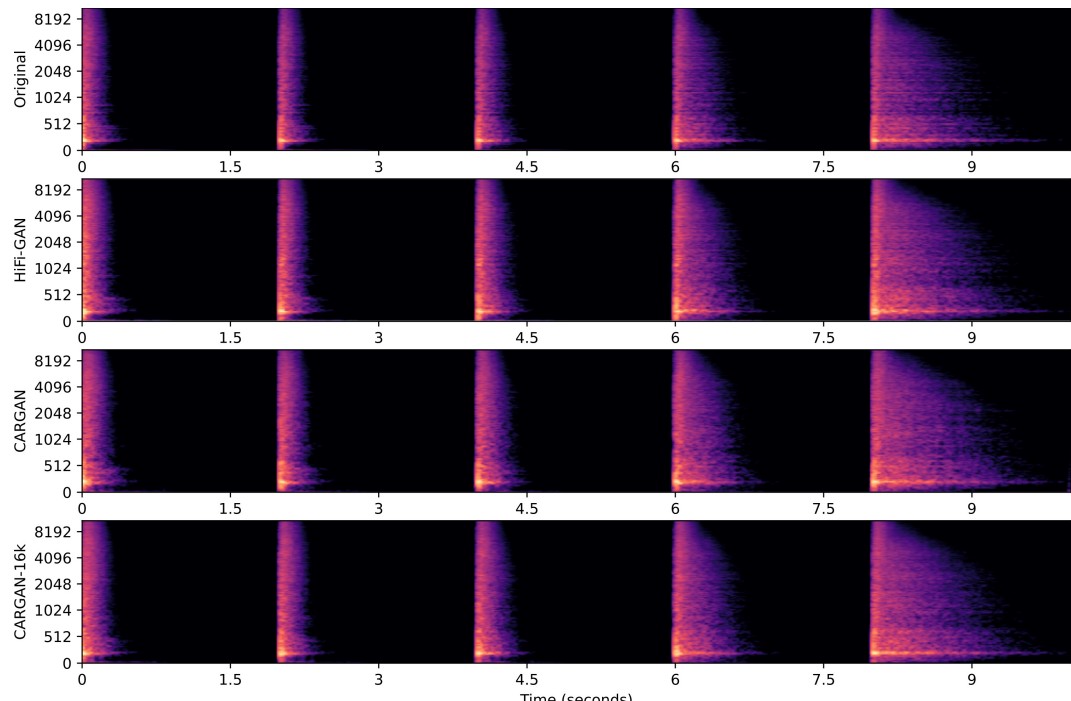

Figure 6: Spectrogram visualizations for our reverb modeling experiment on a repeated snare sample with a gradually increasing reverb decay time. All models are capable of reproducing reverb in this simple case. Comparing the high-frequencies of each transient shows that CARGAN-16k is most capable of accurately modeling the high-frequencies.

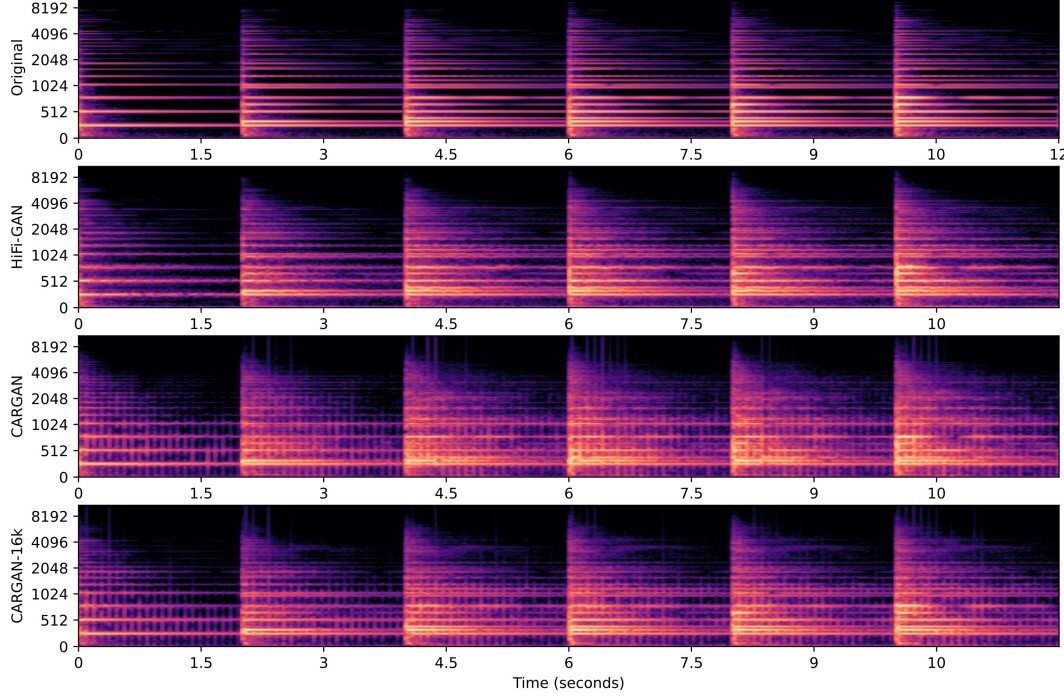

Figure 7: Spectrogram visualizations for our polyphonic modeling experiment on a MIDI piano instrument with a gradually increasing number of notes. HiFi-GAN exhibits a strong vibrato that is easier to hear than to see on a spectrogram. CARGAN and CARGAN-16k exhibit periodic boundary artifacts which are clearly visible in the spectrogram.

