# OpenReview forum: "Chunked Autoregressive GAN for Conditional Waveform Synthesis"
_ICLR.cc/2022/Conference — ICLR 2022 Poster_

### Official Review · Reviewer_Zbrk · 2021-10-17

**Correctness:** 3
**Technical Novelty And Significance:** 2
**Empirical Novelty And Significance:** 3
**Recommendation:** 6
**Confidence:** 4

**Main Review:**

This paper proposes a novel conditional waveform generative model, CARGAN, combining the advantages of AR and non-AR models.
Furthermore, it not only improves the performance of the CWS, but it also conducts the side-by-side listening comparison and pitch analysis, which makes it easier for readers to understand what is the problem of the previous non-AR CWS model. Also, I think that the consideration about choosing the optimum chunk size depending on the size of the receptive field also can give good insights to future works.

However, I have several concerns about this work.

First, I am not sure that the comparison of the ability to learn the cumulative sum operator is enough to explain the performance gap. This is because, unlike the cumulative sum operator, there is a range for the phase values ([0~2pi]), so it might be possible for HIFI-GAN to cover the phase if it has enough receptive field to cover the phase. If it is not, I think it would be better if there is an experiment showing the change of the performance when the length of the input gets longer, because the loss for learning cumsum gets larger when generating full sequence.

Second, although it conducts diverse analyses and can give many insights about designing the architecture of the model, I think there is a slight lack of novelty. As mentioned in Section 1, there have been several models that generate audio autoregressively in the chunk level [1]. Also, in text-to-speech, there is already a technique of using a reduction factor that controls the trade-off between the sampling speed and the speech quality [2].

Third, there are several questions about the paper that I am confused about.
* In Section 2, it is a little difficult for me to understand how penalizing the L1 distance between activations leads to requiring a model to produce a waveform that is close to the ground truth waveform.
* According to Figure 2, it seems that a part of the generated speech is fed back to the model. Is it also true during training or the ground-truth segment is used? (i.e. teacher-forcing)
* If the CARGAN and HIFI-GAN have the same generator and discriminator architecture, how the training speed and memory usage can be improved? I think CARGAN would rather be slightly inferior because of the additional previous waveform information. If this is because of the difference in the sequence length used in training (8192 vs. 2048), what happens when training the HIFI-GAN with 2048 segment size?

Lastly, there are some personal opinions about this paper.
* As far as I know, there is a method to implement the causal convolution with the non-causal convolution layer based on padding operation [3]. Therefore, I think the maximum length of a learnable cumulative sum is not [(m+1)/2].
* I think it would be better if the loss is described more in detail to understand the loss compensation technique.
* I wondered about which chunk size 'k' is used in the experiment of training the cumulative sum operation, in order to understand the relationship between receptive field, chunk size, and the sequence length.
* There is no subsubsection 5.2.2, so I think the subsubsection 5.2.1 can be combined with subsection 5.2
* I think that it would be better if the transcripts for the audio samples in the demo page are written with the words to focus emphasized.


[1] Weiss, R. J., Skerry-Ryan, R., Battenberg, E., Mariooryad, S., and Kingma, D. P. Wave-Tacotron: SpectrogramFree End-to-End Text-to-Speech Synthesis. In ICASSP 2021-2021 IEEE International Conference on Acoustics, Speech and Signal Processing (ICASSP), 2021.

[2] Wang, Y., Skerry-Ryan, R., Stanton, D., Wu, Y., Weiss, R. J., Jaitly, N., Yang, Z., Xiao, Y., Chen, Z., Bengio, S., Le, Q., Agiomyrgiannakis, Y., Clark, R., and Saurous, R. A. Tacotron: Towards end-to-end speech synthesis. In Interspeech, 2017.

[3] https://theblog.github.io/post/convolution-in-autoregressive-neural-networks/

**Summary Of The Paper:**

This paper proposes a conditional waveform synthesis (CWS) model called Chunked Autoregressive GAN (CARGAN). By combining the advantages of the AR and non-AR generative models, CARGAN achieves better pitch accuracy with faster training speed and less memory usage without much decline in generation speed.

First, it shows that the previous GAN-based non-AR CWS models do not accurately preserve the pitch and periodicity of the audio signal. Then, this paper demonstrates that there is a close relationship between pitch and phase, and the inductive bias of AR model is good at learning the relationship. Especially, this paper supports the claim by conducting an experiment that compares the ability of  AR and non-AR models to learn the cumulative sum operation, because the instantaneous frequency and instantaneous phase have the cumulative sum relationship. Lastly, by comparing HIFI-GAN and CARGAN based on spectrogram-to-waveform conversion task, this paper shows that CARGAN reconstructs the original waveform more precisely. Furthermore, CARGAN shows faster training speed and less memory usage than those of the HIFI-GAN.

**Summary Of The Review:**

Overall, I think this paper gives several insights to design a conditional waveform synthesis model by combining AR and non-AR TTS model.
However, I think it slightly lacks novelty in that there have been several models designing an autoregressive model with chunk-level output. Also, I think it is a little weak in supporting its claim and there is room for improving the writing. As a result, I will give a score of 5 for this work.

---

> ### Author Response · Authors · 2021-11-20
> **Response to Reviewer Zbrk**
>
> We thank you for your thoughtful and detailed feedback on our paper. We appreciate the questions and concerns that you have raised. We hope that our individual responses below will help resolve some of those concerns.
>
> **Q1: First, I am not sure that...**
>
> We do not claim that performance on the cumulative sum operator fully explains the performance gap. The cumulative sum task exposes strictly different behaviors between non-autoregressive generators, autoregressive generators with causal receptive fields less than the chunk size, and autoregressive generators with sufficiently large causal receptive fields. While phase values are bounded to the range [0, 2pi], our cumulative sum task is bounded to [0, 1]. We agree that HiFi-GAN could be improved if its receptive field could cover the phase, but it’s unclear how to do this in the non-autoregressive case for audio clips of millions of samples or more. The cumulative sum task demonstrates that autoregressive models can cover the phase, but only when the causal receptive field is sufficiently large. For example, we mention in Section 4.2 that auxiliary phase metrics were reduced when using autoregression. These improvements were typically at least one order of magnitude for up to a few chunks, and reduced thereafter. As with the cumulative sum task, the longer the sequence is the larger the phase error becomes. We hypothesize that this is due to two causes: (1) small errors in phase prediction compounding autoregressively and (2) certain regions of high-entropy phase (e.g., unvoiced sounds) cause the starting phase of the next voiced region to be non-deterministic. So we believe that there are factors other than length that impact the accuracy of the phase modeling. Exploring those factors is an interesting direction for future work.
>
> **Q2: Second, although it conducts...**
>
> Wave-Tacotron should be cited. Thanks for catching this. This has been added to Section 1 under “Hybrid methods”. We do not claim as a novel contribution the ability to trade-off between sampling speed and speech quality (although we do show trade-offs between sampling speed + {pitch accuracy, periodicity accuracy, training speed}). As well, the reduction factor technique of Tacotron is specific to attention-based alignment models with a low output temporal resolution (e.g., spectrograms) and is not applicable in, e.g., the spectrogram-to-waveform inversion task.
>
> **Q3: In Section 2, it is...**
>
> Consider the activations after the first convolution layer + non-linearity. The convolution is a learned filter bank that makes modified copies of the signal; it might modify the phase, but deterministically for both generated and ground truth. The non-linearity can destroy the phase information of signals without zero-crossings, but typically our signals have zero-crossings. So the phase is still in-tact at the first point where activations are compared between generated and ground truth audio. Non-autoregressive generators lack the pitch + phase information to accurately generate the rest of this phase. This phase mismatch causes the large losses of these early discriminator layers to dominate training.
>
> **Q4: According to Figure 2, it seems...**
>
> We experimented with using Unrolled GANs [1] to, as you mention, train on generated speech. We found that this induced significantly worse artifacts.
>
> [1] Metz, L., Poole, B., Pfau, D., & Sohl-Dickstein, J. "Unrolled generative adversarial networks." ICLR 2017.
>
> **Q5: If the CARGAN and HIFI-GAN have the same...**
>
> You are correct that the speed difference is caused by the sequence length used in training. Our experiments with HiFi-GAN and a smaller segment length produced audio with worse subjective quality to our ears.
>
> **Q6: As far as I know, there is a method to implement the causal convolution...**
>
> The padding described is a common method of creating a causal convolution from a non-causal convolution. Our analysis applies only to non-causal convolutions without the padding that you mention, which are currently the more common choice within audio generator architectures.
>
> **Q7: I think it would be better if the loss is described more in detail**
>
> We added a few sentences at the start of Section 4.2 that provide details relevant to the loss compensation technique.
>
> **Q8: I wondered about which chunk size 'k' is used...**
>
> Thank you for catching this missing detail. The chunk size and autoregressive conditioning size are the same as for spectrogram-to-waveform inversion of speech. We have added this information to Section 5.1.
>
> **Q9: There is no subsubsection 5.2.2...**
>
> We have combined these subsections
>
> **Q10: I think that it would be better if the transcripts...**
>
> We assume this suggestion is to replace the phrases “Listen for the artifact on the word X” with the transcript itself, with X bolded. We have updated our companion website accordingly. This will take effect on the deanonymized website once that is released.

---

> > ### Comment · Reviewer_Zbrk · 2021-11-23
> > **Response to the authors**
> >
> > Thank you for your considerate responses to my review and it has resolved many of the questions, and also I appreciate you for reflecting my several opinions in the revision. I also hope the answers can be also reflected in future revisions. (e.g. how the memory consumption gain can be obtained.)
> >
> > In addition, I have some minor opinions that might be done in the paper.
> > * In Section 5.1, what it will be like if HIFI-GAN is trained with the larger kernel setting? (larger causal receptive field than the signal lengths.)
> > * I think it would be better if it can be clarified that the original HIFI-GAN is trained to convert a mel-spectrogram segment of length 32, but it is used to convert a much longer mel-spectrogram.
> >
> > As a result, I will increase my score to '6'

---

> > > ### Author Response · Authors · 2021-11-23
> > > **Response to Reviewer Zbrk**
> > >
> > > Thank you for increasing your score! We are glad we were able to resolve your questions.
> > >
> > > A large kernel HiFi-GAN is an interesting case to consider. One of the reasons we don't further pursue the large kernel models in this paper is that they incur large (linear) costs in generation speed and memory consumption. For HiFi-GAN, this is a 34x degradation in generation speed and memory consumption. We believe that further attention should be devoted to increasing the receptive field of the generator without incurring these costs. The GAN-TTS architecture is a step in this direction, using large strides and dilations, but still incurs a 21x cost. The memory consumption in particular makes these experiments difficult.
> > >
> > > In the case where the causal receptive field of a non-autoregressive generator is larger than the training length $\ell$, our hypothesis is that the cumulative sum output for long inputs will be correct over the first $\ell$ samples. After that, it will be a cumulative sum over the last $\ell$ samples. So the graph will look linear for the first $\ell$ samples and afterwards constant.
> > >
> > > The rebuttal submission deadline has passed, but for future revisions, we will add the following:
> > >
> > > **Section 5.2**
> > >
> > > Add the training length of HiFi-GAN and note that inference length is much longer
> > >
> > > **Section 5.3**
> > >
> > > Add that the memory improvements during training are primarily due to changes in training sequence length $\ell$

---

### Official Review · Reviewer_agkP · 2021-11-02

**Correctness:** 4
**Technical Novelty And Significance:** 4
**Empirical Novelty And Significance:** 4
**Recommendation:** 8
**Confidence:** 4

**Details Of Ethics Concerns:**

no concerns

**Main Review:**

Strengths:
- the pitch motivation is well presented and natural.
- the solution proposed is a good trade-off between accuracy and speed, and is straightforward to put in place.
- the authors provide extensive evaluations, including MOS.
- the authors provide extensive examples to highlight different types of artifacts

Weaknesses:
- Some remarks (see after) should be addressed.
- a single baseline (HiFi GAN), while their exist a considerable number of vocoders (auto-regressive, flow etc).
- the phase prediction issue is intuitive and could be more compactly treated.


Questions and Remarks:
- In [1], the authors condition on a melody synthetize from the pitch (i.e. a cosine waveform with consistent phase and the right pitch). This means the generator can use the phase from the waveform as a base for its generation. This is much more efficient than giving the pitch directly for the reasons the authors state. This is related to your work, and it could be interesting to see how HiFi-GAN performs with that simple fix. Of course the advantage of the method proposed here is that there is no need to extract pitch information for generation.
- for the toy task, the results for HiFi GAN should be the same pattern repeated as the input to the model is constant and so each output frame should be the same no?
- can you comment and add speed comparison with other methods, in particular waveglow, wavernn etc.
- do the authors know if Parallel WaveNet would suffer from pitch artifacts ? because it is no longer autoregressive and only conditioned on random noise. intuitively one might think that this would be problematic. Same question for flow based methods like waveglow.

There a number of typos:
- p2. 'deniosing'
- p2 'hybird'
- p 4: 'Autoregression is an sensible' -> 'a sensible'


[1]: Unsupervised Cross-Domain Singing Voice Conversion, Polyak et al 2020.

**Summary Of The Paper:**

The authors present a chunk-wise auto-regressive generative model for audio with adversarial loss.
In particular, the authors note the limitations of purely convolutional adversarial audio generation model for text to speech as those fail to provide a consistent pitch for an extended duration. The authors provide a number of audio samples to illustrate the issue.
They evaluate the proposed method on standard datasets (VCTK) and perform subjective evaluation. they show that their method achieve better pitch correctness, and can improve perceptual evaluations, in some cases.
They also evaluate the speed of evaluation and training of their method. Unsurprisingly, the method proposed is slower than non auto-regressive generation but the chunking makes it theoretically faster than purely auto-regressive methods.


**Summary Of The Review:**

Simple method suggested by the author to fix some known artifacts from GAN audio vocoders.

---

> ### Author Response · Authors · 2021-11-20
> **Response to Reviewer agkP**
>
> Thank you for your thoughtful and detailed feedback on our paper. We appreciate the interesting questions, and respond to each question separately below.
>
> **Q1: In [1], the authors condition on a melody synthetize from the pitch (i.e. a cosine waveform with consistent phase and the right pitch). This means the generator can use the phase from the waveform as a base for its generation. This is much more efficient than giving the pitch directly for the reasons the authors state. This is related to your work, and it could be interesting to see how HiFi-GAN performs with that simple fix. Of course the advantage of the method proposed here is that there is no need to extract pitch information for generation.**
>
> We believe the described method of using a source-filter method with a predetermined signal is very relevant, and is most similar to HooliGAN [2]. We agree that this method currently requires highly accurate pitch extraction, as phase error will compound autoregressively when the pitch or voiced/unvoiced classification is incorrect. As well, this method limits us to the generation of monophonic audio data. However, we hypothesize that this method can also be used in conjunction with HiFi-GAN to improve speech quality. The public demo of HooliGAN (https://resemble-ai.github.io/hooligan_demo/) does not seem to exhibit the pitch or periodicity artifacts addressed in our paper.
>
> [1] Unsupervised Cross-Domain Singing Voice Conversion, Polyak et al 2020.
>
> [2] McCarthy, O., & Ahmed, Z. (2020). HooliGAN: Robust, high quality neural vocoding. arXiv preprint arXiv:2008.02493.
>
> **Q2: for the toy task, the results for HiFi GAN should be the same pattern repeated as the input to the model is constant and so each output frame should be the same no?**
>
> We considered this variant, where HiFi-GAN is fed the ground-truth running sum and generates in chunks of 8192. However, we do not include it as this becomes an autoregressive model with a fixed chunk size of 8192, whereas HiFi-GAN can generate many more than 8192 samples in parallel. So this proposed variant is no longer a non-autoregressive HiFi-GAN, which is our intended baseline.
>
> **Q3: can you comment and add speed comparison with other methods, in particular waveglow, wavernn etc.**
>
> We added an appendix (Appendix I) that compares the speed of CARGAN and HiFi-GAN to prior methods using measurements collected from prior works.
>
> **Q4: do the authors know if Parallel WaveNet would suffer from pitch artifacts ? because it is no longer autoregressive and only conditioned on random noise. intuitively one might think that this would be problematic. Same question for flow based methods like waveglow.**
>
> WaveGlow exhibits audible pitch errors that resemble the ones that our method addresses. In the public demo of WaveGlow (https://nv-adlr.github.io/WaveGlow), listen for the word “form” in the second-to-last and the word “used” in the last example of spectrogram-to-waveform inversion. Parallel WaveNet is an interesting case, as the teacher network is autoregressive and does not exhibit these artifacts. We hypothesize one of two possibilities: (1) the autoregressive teacher of Parallel WaveNet can resolve these pitch issues (even with a non-autoregressive generator), while the non-autoregressive discriminator of HiFi-GAN cannot or (2) the teacher of Parallel WaveNet is trained with pitch conditioning, which provides comparable benefit to pitch conditioning the discriminator, as described in Appendix B. I could only find a few officially released examples from Parallel WaveNet (https://deepmind.com/blog/article/wavenet-launches-google-assistant). We could not hear any of the pitch or periodicity artifacts described in this paper, but it is difficult to tell without corresponding ground truth. The current state-of-the-art in pitch accuracy is CLPCNet (https://arxiv.org/abs/2110.02360), a paper currently under review that exhibits slightly greater pitch accuracy (in terms of RMS) than CARGAN.

---

> > ### Comment · Reviewer_agkP · 2021-11-22
> > **reply to authors**
> >
> > Thank you for your reply and adding the speed comparison.
> >
> > For the comparisons with methods like [1] and [2], I think it could be good to add it to the introduction, with the mentioned limitation (F0 estimate + monophonic).
> >
> > For Q2, i wasn't asking to train HiFi GAN with auto-regression,  but why it doesn't repeat its output pattern. The reason might be because of the overlap between frames, and only the rightmost and leftmost frames have no overlap with another frame and can predict the proper answer. For instance, changing the transposed convolutions to have zero overlap (stride = kernel size) would likely make a repeated pattern appear.

---

> > > ### Author Response · Authors · 2021-11-22
> > > **Response to Reviewer agkP**
> > >
> > > We agree that adding source-filter models to the introduction is beneficial. We have added Neural Source-Filter [3], a canonical source-filter model that uses the same principle (generate a deterministic periodic waveform and use a neural net to transform it to speech) and has the same limitations.
> > >
> > > Thanks for clarifying Q2. If you look closely on the left-most plot of Figure 3 (cumsum with a chunk size of 1024), you'll see that HiFi-GAN transitions to a horizontal line at roughly 245 (i.e., the size of its causal receptive field) and 1024 - 245. The convolution layers of HiFi-GAN are zero-padded. So we believe the network is using the zero-padding as location markers, and then filling in values using the average slope until it reaches the end of the causal (and anti-causal) receptive field. Once you're past the end of the unidirectional receptive field, the best the network can do is predict the average that it saw during training. This hypothesis explains the observed behavior in HiFi-GAN as the chunk size increases, as well as why a repeated pattern is not expected.

---

### Official Review · Reviewer_dhJa · 2021-11-03

**Correctness:** 4
**Technical Novelty And Significance:** 3
**Empirical Novelty And Significance:** 4
**Recommendation:** 8
**Confidence:** 4

**Main Review:**

Strengths:
1. The paper is well presented and the reasoning is well discussed.
2. Literature review is satisfactory, but some more references can be added.
3. The artifacts caused by parallel generation are highlighted in the web page with extensive audio examples.
4. The method is evaluated by looking at pitch, periodicity errors and F1 score of voiced/unvoiced classification as well as A/B comparison subjective listening tests with other conditional synthesizer methods.

Weaknesses:
1. An example of hybrid methods would be: "Wave-tacotron" which is very similar to this paper but uses flow instead of GANs. I think it needs to be cited.
2. The boundary errors are quite audible in music examples even with context inclusion. Maybe the hyper-parameters such as chunk size and context size need to be tuned for music.

**Summary Of The Paper:**

The paper introduces a Chunked Autoregressive GAN (CARGAN) method for conditional synthesis which is autoregressive over chunks of audio but uses Hifi-GAN like parallel generation within a chunk. The method is motivated by the periodicity and pitch errors shown by existing parallel (non-autoregressive) GAN generators. The authors argue that the periodicity and pitch errors are caused by the parallel GAN generators which may disregard continuity of the periodicity and pitch when generating audio. The auto-regression makes sure that the generated audio is less error-prone in regards to continuity and accuracy of the periodicity and the pitch. The discriminator takes in some contextual samples as well as the chunk that is generated to make sure there is less boundary artifacts.

**Summary Of The Review:**

The paper reads well and has a very extensive demo page. I believe it furthers the state of the art in this area.

---

> ### Author Response · Authors · 2021-11-20
> **Response to Reviewer dhJa**
>
> We thank you for your feedback and recommendations for improvement. Each of the weaknesses you indicated are addressed separately below.
>
> **Weakness 1: An example of hybrid methods would be: "Wave-tacotron" which is very similar to this paper but uses flow instead of GANs. I think it needs to be cited.**
>
> Thanks for noticing this. We have updated the manuscript to include Wave-Tacotron as an additional hybrid method.
>
> **Weakness 2: The boundary errors are quite audible in music examples even with context inclusion. Maybe the hyper-parameters such as chunk size and context size need to be tuned for music.**
>
> We agree the boundary artifacts are audible on some musical examples. We tried both large (16384) and small (512) context sizes without substantial improvement to boundary artifacts. We have not tried a chunk size ablation on music. On speech, increasing the chunk size reduces boundary artifacts in quantity (i.e., fewer boundaries), but not in severity. We have two additional hypotheses regarding the existence of these boundary artifacts: (1) The AR conditioning is out-of-domain. Specifically, there is a mismatch between the generated and real audio used as autoregressive conditioning, as CARGAN is never trained using generated audio as conditioning, and (2) The input spectrogram is out-of-domain. Music is a much higher-entropy distribution than clean, non-reverberant speech, which makes generalization difficult without much higher capacity models, such as Jukebox [1]. This would explain why the boundary artifacts are much worse on music than speech. For (1), we attempted using unrolled GAN [2] to train the model on its generated output, but this induced worse artifacts.
>
> [1] Dhariwal, P., Jun, H., Payne, C., Kim, J. W., Radford, A., & Sutskever, I. (2020). Jukebox: A generative model for music. arXiv preprint arXiv:2005.00341.
> [2] Metz, L., Poole, B., Pfau, D., & Sohl-Dickstein, J. "Unrolled generative adversarial networks." International Conference on Learning Representations (ICLR). April 2017.

---

> > ### Comment · Reviewer_dhJa · 2021-11-30
> > **Response to authors**
> >
> > I read the response by the authors and the other reviews.
> >
> > Thanks for the response.
> >
> > I am satisfied with the changes to the paper.

---

### Author Response · Authors · 2021-11-20
**Summary of Paper Changes**

We thank all of the reviewers for their time and feedback in improving our paper. Here is a summary of the changes that have been made in each section.

**Section 1**
- We cite and describe Wave-Tacotron as an example of a hybrid method.

**Section 4.2**
- We add two sentences at the beginning of the section to provide additional details about the loss functions that are relevant to the section.

**Section 5.1**
- We add the second sentence in the fourth paragraph of this section ("We train CARGAN with...") describing the hyperparameters used during the cumsum experiment.

**Section 5.2**
- We merge sections 5.2 and 5.2.1.

**Appendix I**
- We add an appendix that provides speed comparisons with other recent models in the literature.

We also updated our website to contain more detailed transcripts (this will take effect when our deanonymized website is released) and fixed several typos throughout (thanks to reviewer agkP).

---

> ### Author Response · Authors · 2021-11-22
> **Additional paper change**
>
> We have made the following change based on additional discussion with Reviewer agkP.
>
> **Section 1**
> - We cite and describe source-filter models (specifically the Neural Source Filter model) as an additional example of non-autoregressive models

---

### Decision · Program_Chairs · 2022-01-20

**Decision:**

Accept (Poster)

**Comment:**

This work proposes a hybrid autoregressive and adversarial model for sound synthesis (including but not limited to speech), conditioned on various types of control signals. Although recent adversarial approaches have gained favor over previously popular autoregressive approaches in this domain, because of their ability to produce audio signals much more quickly, the authors argue that these models tend to introduce certain types of artifacts which stem from an inability to learn accurate pitch and periodicity. They propose to address this by reintroducing some degree of autoregression, without compromising too much on inference speed.

Reviewers praised the presentation of this work, the thoroughness of the experimental evaluation, and the audio examples provided. A few concerns were also raised regarding related work and the clarity of some parts of the paper, which the authors have taken the time to address. After the discussion phase, all reviewers chose to recommend acceptance, and I will follow their recommendation.